# The Room: design and embodiment of spaces as social beings

## ABSTRACT

This paper delves into the exploration of spaces as non-anthropomorphic avatars. We are investigating interaction with entities showing features different from humans', to understand how they can be embodied as avatars and perceived as living, social beings. To push this investigation to its limit, we have designed as an avatar an interactive space (the Room), that challenges both the anthropomorphic structure, and most of the social interaction mechanisms we are used to. We introduce a pilot framework for the Room design, addressing challenges related to its body, perception, and interaction process. We present an implementation of the framework as an interactive installation, namely a real-time, two-player, VR experience, featuring the Room avatar, with a focus on haptic feedback as the main means of perception for the subject embodying the Room. By radically challenging anthropomorphism, we seek to investigate the most basic aspects of embodiment and social cognition.

## CCS CONCEPTS

• **Human-centered computing** → **Virtual reality**; **Haptic devices**; *Empirical studies in HCI*; *User interface design*; • **Applied computing** → *Media arts*.

## KEYWORDS

Interactive Spaces, Interactive Installations, Multi-Modal Interfaces

## 1 INTRODUCTION

Imagine you could step inside a room that is more than just a lifeless space, designed to interact with you and communicate with you through movement and other non-verbal cues, an enclosed space that has come to life. This project is part of a research investigating "non-anthropomorphic avatars" and consists of the design and implementation of spaces that are avatars to be embodied, capable of engaging and communicating with human visitors inside them.

We are particularly interested in how new technologies give the possibility to explore bodies beyond our own anthropomorphic shape [16, 44], challenging the fundamental notions of what makes a body alive and social.

Recent explorations are addressing the topic of embodying non-anthropomorphic avatars [22, 29, 41]. Can we seamlessly enter bodies that are non-humanoid? Researches on embodied cognition [12, 37, 51, 56] assign to the body a central role in the way we process and understand reality. Thus, exploring the possibility to radically alter our bodies supports the exploration on the possibility to go beyond the current understanding of our cognition.

We investigate what allows a person to perceive another, non-anthropomorphic being, as "alive". What sets of physical elements, movements, sounds, actions create "the illusion of life"? This research question has applications ranging from social robotics (*"how can machines seamlessly be integrated in our social tissue?"*) to entertainment (*"how to design characters that can be understood and empathised with?"*).

To focus our research on a specific type of body, as radically different from the human structure as possible, we chose organic, enclosed spaces. We call these types of bodies *the Room*.

### 1.1 The Room as an Avatar

Animals, fictional characters, social robots and humans all have fundamental elements in common:

- They are "points in space", with a concept of personal space.
- They can move in space. This allows them to control their personal space and to interact with the environment, which is something "other" than them.
- They have a "face", the location of most of our senses, and the spatially limited focal point of interaction.

The Room disrupts all these elements at once: it is both a participant and the setting for the interaction, making the relationship with the human partner asymmetric: one being entirely contains the other. Concepts of personal space, attention and perception are lost or dramatically changed.

### 1.2 The Setup: Interaction Through the Digital Filter

Our interaction setup features two human subjects, interacting with each other in *Virtual Reality*:

- The **Controller** embodies the virtual non-anthropomorphic avatar, the Room.
- The **Visitor** interacts with the non-anthropomorphic avatar, by entering the Room in VR.

Therefore, the interaction between users is mediated by a *"digital filter"*, to such an extent that they may not be co-present or aware of each other's physical attributes (age, voice, shape, ..) or human essence.

All communication is non-verbal, since verbal communication plays a very important role in human-human communication and, when possible, humans will prefer it as the main communication medium. Removing verbal communication forces participants to explore and discover creative ways to utilize the *digital filter* to establish a shared communication code.

### 1.3 The Contribution

We present three contributions.

First, the formulation of a *pilot framework* for the design of the elements of the Room experience.

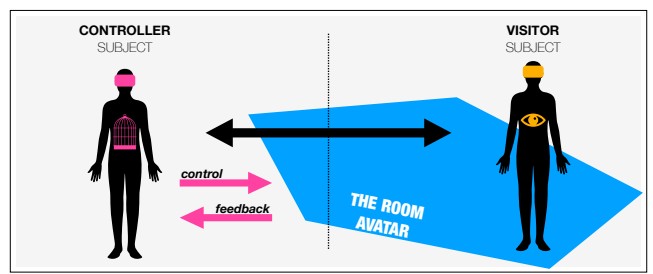

**Figure 1: Diagram of the interaction in *the Room* setup. Two users interact non-verbally: the Visitor subject enters an animated space, the Room, while the Controller subject embodies and animates (control and perception) the Room, its avatar.**

The second contribution is the implementation of a first version of this system, in the form of an *interactive installation* presented to the public: a real-time, two-player VR experience, also used to gather data.

Touch is the human sense most coherent with the perception of the Room, a distributed being. Consequently, our third contribution involves exploring and developing *haptic devices* integrated into virtual reality, emphasising sensations, wearability, and implementation, to create the sensory translation system. The relevance of this exploration extends beyond the scope of this paper, shedding light on the broader application of technology to enhance touch, an underutilised sensory channel in virtual reality.

Overall, this paper is a pilot study of how multimodal interfaces (haptics and VR) can be leveraged to embody and interact with non-anthropomorphic avatars, and in particular, spaces.

The paper is organised as follows. *Section 2* presents a background for this research, which is the foundation for the framework that is presented in *Section 3*. In *Section 4* we introduce the haptic devices we implemented. *Section 5* describes the Interactive Installation we developed. *Section 6* contains a discussion on the experiential results. *Section 7* concludes the paper with reflections and ideas for future developments.

## 2 BACKGROUND

Here we provide a comprehensive research that addresses our two core problems: embodying non-anthopomorphic avatars and creating living, interactive spaces.

### 2.1 Avatar Embodiment

"The Sense of Embodiment (SoE) toward a body B is the sense that emerges when B's properties are processed as if they were the properties of one's own biological body." [38] The term is used to refer to the set of sensations that arise in conjunction with being inside, having, and controlling a body.

*2.1.1 Non Anthropomorphic Avatars.* Results show that body ownership [17, 38] can be felt for bodies or limbs less morphologically similar to the human [10, 31, 39, 41, 51], possibility based on perceived similarity of their functionality [20]. Minimal representations of the body were found to be the most recognizable by users [78] and resulted in increased exploratory behaviors and creativity without lowering the sense of embodiment [45, 76]. In [31], users also indicated sense of agency, the feeling that one is causing or generating an action [38], for point light representations of their limbs when these were coherent with their real movements. Participants can identify themselves with, and control, avatars with different morphology than the human's [41, 53, 72, 80]; synchronous visual-motor control was found to be a necessary and sufficient condition for embodiment. However, strong unnatural relationships to more human-like visual cues may be detrimental for SoE and a more abstract representation of the avatar may increase the sense of ownership and performance [66, 79]. Users can be immersed in their non-anthropomorphic avatars by forcing the new morphology on the human body [22] or adding a feedback on the control actions [29].

*2.1.2 Sensory Alterations.* Inhabiting a new body also means choosing different forms of perception, and hence changing our understanding of the world. For example, visual information can be transformed into tactile stimuli or auditory signals can be visualized [11, 58]. The Reality Helmet [77] is an example of altered embodiment where technology becomes a part of the body and changes the form of perception. In [29], information from the robotic avatar is translated in real-time into minimal, abstract representations in virtual reality to distill affordances and objectives of the new bodies.

*2.1.3 Haptic Technologies.* In recent years, haptic suits, such as the The Bhaptic Suit [15] and Teslasuit [73], have entered the market, designed for integration with virtual reality environments, providing users with a high level of tactile feedback. Experiments such as [69, 74, 81] are attempts to reproduce different types of haptic feelings, but still suffer from the specificity of the application [40], by being too body-specific [69, 74], or poorly portable [81].

### 2.2 Interaction With Spaces

To understand how spaces can be expressive agents within an interaction process, we use principles of Interactive Exhibition Design, with insights on what drives visitors through spaces and how to stimulate them, and an extensive study of existing spaces designed to be interactive installations.

*2.2.1 Interactive Exhibition Design.* The aim of immersive experiences is to engage all senses, extending beyond simply conveying information, to evoke emotion and captivate visitors. Sensory design plays a central role, encouraging interaction and potentially going beyond the conventional five senses: pain (*Nocioception* [13]), balance (*Equilibrioception* [30]), body awareness (*Proprioception* [35]) and sensibility to temperature (*Thermoception* [27]).

Lights and sounds are powerful tools, creating coherent environments [52], illusions [62], giving visitors the possibility to control the space [52, 60], distorting familiar landscapes [33] or creating new ones [75].

Crucial to the experience is routing. In exhibition design theory, a visitor's route is the way he passes through and interacts with the environment. Paths can be linear, with an incremental development [14], or fragmented, granting to the visitors more freedom on the route to follow with many subspace connections [55]. Linear

paths can be easier to experience allowing to focus on the content, but can be considered as simple and dull. High fragmentation and connectivity is empowering and challenging, but can be overwhelming. In F/EEL [50], routing becomes the essence of the installation itself, with obstacles meant to challenge visitors and forcing them to interact with the scenery.

*2.2.2 Spaces as Installations.* Relevant to our investigation are spaces that become living organisms. In [6, 25], organic, sinuous forms and coherent and soft materials like cloth or synthetic hair enveloping the entire space contribute to this global feeling. *Breathing Room* makes the space alive by creating a distributed, mechanised breathing movement [14]. Particularly worth noticing are artists Ernesto Neto [8] with his large, soft, biomorphic installations that invite viewers to touch, poke, and walk on or through, and Rebecca Louise Law [46], leveraging materials from nature to reproduce the feeling of being cocooned in a womb.

The act of moving through the environment is a crucial aspect. Many installations focus on the experience of crossing a border [1, 3], physicalise space [70], create maze-like pathways with walls, curtains or columns [5] or actively narrowing spaces to make exploration more difficult [19]. The space can be filled to emphasise empty regions, conveying a sense of emptiness and loneliness [54]. Bruce Neuman [57] creates spaces difficult to move through, challenging visitors and rewarding them with wider or hidden spaces at the end of asphyxiating corridors.

A different way to influence how the space is explored is by acting on the boundaries. Inflatable, soft or irregular floors and walls [2, 32, 42, 67] challenge the visitors to stay in equilibrium and invites them to play or relax. Very low ceilings [28, 55] force or invite visitors to crouch, crawl or lie down. In [48], the combination of soft floor with Rubens' paintings on the ceiling created an arch of the experience starting from pure play to lying down in contemplation as the ceiling was "discovered" by the visitors looking up. The structures by Toshiko Horiuchi MacAdam [59] are intriguing to adults and children because they are so interactive, fostering play and allowing touch and experience with all senses.

Crucially, all these installations, as they make the act of moving different from the usual, challenge visitors to use their body in novel ways or to experience spaces differently, also challenging their haptic senses, forcing or inviting them to touch or be touched or be enveloped by spaces.

Responsive ambients [4, 43, 49] foster engagement and provoke experimentation: the participant becomes the co-author of this shared experience. Crucially, these mechanisms respond to the visitors locally but can also offer a global feeling.

To foster total immersion in a space that results clear and coherent, often the same elements are repeated, either locally [71] or throughout the entire space [14, 47].

Many spaces feature focal elements, being them lights in dark ambients [7] or elements completely different from the surroundings [46]. This can stimulate curiosity, drive exploration, and give meaning to the entire structure.

Integrating movement adds a new level of engagement while creating the illusion of life. *I Am Storm* [24] and *Tele-Present Wind* [18] use similar concepts to visualise outdoors natural data. Though their installations are not meant to be explored within, they provide an interesting inspiration for possible active routing mechanisms. Soundscapes by Zimoun [82] feature the incoherent movement of simple shapes to create a cohesive space. Seen in its entirety, one almost feels the space itself expressing its own discomfort and anxiety, as in [61], or induce them in the spectator [23]. Softer movements and aesthetics can instead convey a sense of peace and wonder [65]. U-Ram Choe [36] uses metallic sculptures that often take the shape mesmerizing flora made of acrylic and stainless steel to structure living spaces.

Interestingly, all these installations also feature the repetition of simpler modules, in a coherent concertation of shapes, sounds and movements.

## 3 FRAMEWORK FOR THE ROOM DESIGN

We present here a framework to guide the design of *the Room* system.

This framework focuses on the design of the Room body, and on the possible motivations to drive interaction between the Visitor, a human, and the Room, a living space. The framework also addresses the problem of controlling and embodying the Room, from the Controller point of view.

### 3.1 Room Body Design

This section focuses on designing the Room as a living, social being capable of communication, considering both its physical body and potential actions.

In this context, the elements of the space are not all static, but they are manipulated in real-time by the Controller, serving as instruments for the Room's actions. Similar to how humans extend their arms to grab objects, a Room composed of actuated elements can rearrange its configuration or adjust its lighting. These actions serve both to convey the Room's state and to interact with the Visitor. Additionally, certain parts of the Room's body must also be sensory organs, for instance walls sensing touch from the Visitor.

This section offers insights and principles derived from our extensive research on installation design (*Section 2.2*) to guide the design process. These elements can be combined and adapted to match specific needs.

*3.1.1 Flock of Elements.* Composing the space of multiple similar elements [8, 14, 42] create a feeling of a cohesive, albeit distributed, being, both aesthetically (the elements themselves) and with their coherent movements [18, 24, 65]. When synchronous movements become jittery and desegregated, they convey a feeling of discomfort or attract the Visitor's attention to a specific location [61].

*3.1.2 Actuated Boundaries.* The Room can act on its boundaries, by stretching/folding/compressing/releasing them. These actions can be used to trigger dramatic effects, or to set a general background sensation (e.g. calming wave-like movements) [14]. Moreover, they can create physical challenges (e.g. make it necessary for the Visitor to crawl) [28].

*3.1.3 Light and Shadow.* Playing with the presence or absence of light [26], its movement along the Room [4], rhythm and intensity [52, 75], globally or locally [7]. Lighting specific regions and casting others into shadows is also an immaterial form of space

manipulation, and it could also induce playful behaviours [49]. Pulsating lights are used also to represent breathing [36, 75].

*3.1.4 Sounds.* Sound modulation is an effective tool for expression, as the tone and "colour" of the sounds could be used to represent emotional states. Similarly to lights, the sound is either global, "ambient", or localised [33, 82], even moving in space [60].

*3.1.5 Routing: Space Manipulation.* The Room can manipulate its space to influence the Visitor based on principles outlined in Section 2.2.1:

- Long corridors encourage faster movement and less attention to content.
- Obstacles induce slow movement.
- Doors and narrow passages create distinct spaces, leading visitors to anticipate different content in each area.

Visitor expectations are crucial: spaces that are separated or more difficult to reach [19] increase the expectation of being rewarded [3, 57], either with novel content or with some other form of discovery.

*3.1.6 Thermoception.* Controlling the temperature of the space to either transmit emotions or induce specific feelings or behaviours on the Visitor. For example, making a region much colder may discourage the Visitor from traversing it. This manipulation can be global [21, 27] or local [34].

*3.1.7 Equilibrioception.* By rearranging its shape the Room can influence the sense of equilibrium of the Visitor. Regular, stable surfaces allow the Visitor to retain its usual perception, while moving the floor, or creating incongruent visual arrangements [30] will discombobulate them. This can also be obtained with soft, unstable or actuated floors [42, 48, 67].

*3.1.8 Spatial Attention.* In traditional bodies, we can always tell if someone is focusing on something specific, but the Room lacks concepts of eyes or faces. However, by actuating its body, the space itself, locally [23, 36, 61], it underlines that those locations are relevant. Additionally, this mechanism fosters interactive play by allowing the Room to specifically target or not the Visitor [4, 43].

*3.1.9 Focal Elements.* Room designs may require elements central to the interaction, such as more sensitive areas or elements intended to evoke playful responses from the Visitor. These components should stand out or feel distinct from the surroundings [7, 26, 46], signaling their importance. Hinting at their presence beyond boundaries can also spark curiosity [3]. The principles for directing attention of Section 3.1.8 can all be applied.

*3.1.10 Organic Space.* The Room should feel like a living being, overcoming the bias that spaces are functional part of our lives and not independent organisms. Imperfect boundaries [25], uneven components with different sizes and shapes [42, 61], multiple layers [2, 59], and natural materials [6, 46], bring the Visitor towards an imaginary more related to living beings [8].

*3.1.11 Haptic Interactions.* The Room can engage in various physical interactions with the Visitor, such as hugging, touching, pushing, and pulling. For instance, inflatable columns could hug by inflating around the Visitor or push by rapidly inflating next to her/him [19, 42]. Additionally, elements can be designed to sense these actions from the Visitor onto the Room, such as soft elements capable of detecting different qualities of touch [8, 9, 59].

## 3.2 Room Embodiment

Our goal is to develop a system that immerses subjects in a new body. To achieve this, we must understand the differences between the human body and the avatar body.

- The Room cannot move in space but interacts by having the other inside itself.
- It consists of numerous, distributed elements.
- Unlike humans, it lacks a face and perceives its surroundings through distributed sensing.

A body acts in space and receives back information as sensations. These two channels of interface with reality are now mediated by the Room body, thus these two design problems, that of control and perception, need to be addressed.

*3.2.1 Control.* The control system maps the user's body to the avatar's. We implement *Proprioception Remapping on Control* [29], providing haptic cues based solely on the user's control signals. This decreases the expectation of human body movements, and underlines the actions of the user that are now linked to the new avatar's body.

We gather the possible control schemes into two categories.

*"Puppeteer".* Manipulating devices to drive the new body "from the outside", akin to puppeteering, video games, and driving vehicles. Control mapping and avatar shapes can be very flexible, but the perceived distance between the two bodies leads to lower embodiment.

*"Body-to-Body".* Fully embodying the Room's body by emulating its spatial conditions (for example having the Controller on the floor or lying with the belly on a chair to have its body take the shape of an envelope [22]) and using control devices engaging the entire body. This may enhance embodiment, but requires a complex control system correlating human and Room's morphology.

*3.2.2 Sensory Translation.* Immersing the Controller into a radically new body involves a different perception of reality, which impacts on affordances and on the relationship between actions and effects, e.g., in terms of positive and negative sensations. We aim for the Controller to empathize with the needs of the Room as its own, and act accordingly.

The Sensory Translation system gathers the data from the avatar's perception and translates them into signals that can be perceived by the subject, for example translating the Visitor's touch into pressure transmitted to the Controller through a haptic vest.

To foster the sense of embodiment, stimuli should include representations synchronous to the control signals, and multi-modal, to create a unified source of body ownership [63, 64, 68].

To align with the Room's distributed nature and lack of a face, visual input for the Controller should be minimized, emphasizing haptic sensations instead. While sound shares limitations with vision, it can be used to influence emotional states or increase immersion.

## 3.3 Designing Interaction

It is crucial to foster interaction between the Room and the Visitor, motivating them to overcome their differences and establish a communication code.

*3.3.1 Social Motivations for the Room.* Typically, spaces are designed as functional to humans needs, leading to a bias in envisioning interactions where only the Room serves the Visitor. However, as the Room is as a living, social entity, it must have needs of its own, specifically ones that only the Visitor can fulfill.

Social beings, including the Room, require each other for tasks that cannot be accomplished alone or to streamline processes.

We isolated the following basic drives:

- Perceiving the Visitor
- Persuading the Visitor

*Perceiving the Visitor.* Sensing can be a motivation for the Room. Like blind organisms such as plants or deep-sea creatures, the Room relies on touch as its main sensory channel. Consequently, the Visitor sometimes may not be perceivable by the Room and the room may actuate its sensible parts to look for her/him.

*Persuading the Visitor.* The Room could need the Visitor to:

- Remove a source of discomfort. For example, a "thorn" inserted in a wall (its skin).
- Obtain pleasure. For example, if the Visitor performs specific actions, like stroking a specific element, the Room (the Controller) will obtain a pleasurable sensation transmitted to the Controller through haptic feedback.

The Room needs to understand how to communicate with the Visitor and persuade her/him to perform specific actions.

A second layer would be to add risk and the need for **trust** to this interaction. For example, if the Visitor's actions may cause discomfort (relayed to the Controller through haptic devices), the Room must cautiously discourage undesired behaviour. The Room must trust that the Visitor will interact appropriately to prevent discomfort or pain

The choice of motivations for the Room guide the selection and specific implementation of the elements presented in Section 3.1, as will be shown in Section 5.

*3.3.2 Social Motivations for the Visitor.* The most straightforward motivation for a human to interact with a space is exploration [6, 7]. The more interesting the space, the higher the motivation [55]. Then there is play and challenge. Interactive spaces often provide physical [4, 43] or mental [49] challenges like reaching specific locations [50], solving puzzles, or understanding connections. Mechanisms that hide portions of the Room or place them out of reach can encourage playful interaction [3, 59], while soft or unstable boundaries add challenge [42, 67]. Such boundaries also prompt the Visitor to engage more deeply with the Room by lying down, touching, or seeking connections [28, 48].

## 4 HAPTIC DEVICES

As our sensory translation system is focused on haptic feedback, this Section provides a description of the methodology and implementation of new types of such devices we developed. These

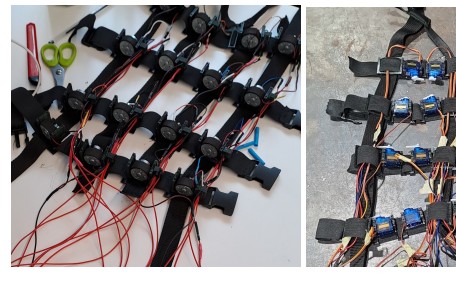

(a) Vibration Suit      (b) Servo-actuated Suit

**Figure 2: Haptic Vests.**

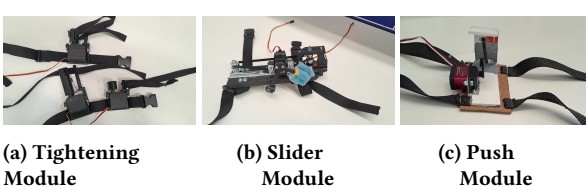

(a) Tightening Module      (b) Slider Module      (c) Push Module

**Figure 3: Haptic Modules.**

devices are not specific to this context and therefore their relevance is general to research in haptic feedback.

## 4.1 Methodology

Investigating spaces as avatars, our objective was to establish communication solely through haptic feedback, minimizing reliance on verbal or visual channels. We aimed to design haptic feedback devices seamlessly integrated with the body's natural sensations, remotely controllable, and capable of providing active feedback. Beginning with a comprehensive list of possible sensations, we discarded those unattainable with high dynamic or compact, wearable devices. From the remaining sensations, we selected a final set to maximize perceived differences between the effects of different devices. Drawing from existing research and the homunculus model, we developed prototypes with clear, recognizable, and user-friendly sensations. Prototypes were iteratively refined based on trials to enhance usability and signal clarity.

## 4.2 The Devices

We designed and implemented four different types of devices, integrating 3D printing with electronics. Their main and common feature is the adaptability to different body shapes and sizes, known to be a limitation for the adoption of haptic devices. Moreover, all but the feedback vests are designed as modules that can be placed seamlessly in different positions of the body.

*4.2.1 Haptic Feedback Vests.* We implemented both a variation of an existing haptic suit based on vibration motors (Figure 2a), and an entirely new servo suit, incorporating servo motors as a different type of actuator (Figure 2b). Our initial focus was on determining the optimal placement of motors on the human torso while ensuring symmetry. Thorough testing helped identify the best orientation for each actuator, and specialized hosting modules were designed

to seamlessly accommodate both types of motors. Each haptic feedback vest is equipped with 16 motors arranged in a 4x4 matrix, with independent control implemented using microcontrollers (ESP32 for Vibro, PCA9685, and ESP32 for Servo). The ServoVest utilizes servo motors with anchor-shaped plastic components to deliver tactile sensations resembling a pinch.

*4.2.2 Tightening Module.* The tightening haptic module in Figure 3a integrates a servo motor with a cylindrical component featuring a central fence. A cylinder perforated along the rotational axis holds a strip securely fixed. As the servo motor rotates, the strip coils around the cylinder, creating a tightening sensation and providing the tactile experience of constriction for the part of the body enveloped by the strip.

*4.2.3 Slider Module.* The slider haptic feedback device in Figure 3b is designed to transmit a wide range of tactile sensations to the user. Many haptic sensations we experience have a relative motion with a significant component that is parallel to the skin's surface, sliding along skin. Our module incorporates a continuously rotating motor to generate the sliding motion of a 180-degree servo motor that enables the transmission of two distinct sensations through a carefully designed mechanical structure. Two fabric types, soft and coarse, have been chosen to differentiate positive and negative tactile experiences.

*4.2.4 Push Module.* The pressure haptic feedback module in Figure 3c features a servo motor integrated with a lever mechanism actuating the motion of a curved surface to produce a controlled pressure on a localized area of the subject's body. This module is securely attached to the body through strings.

## 5 THE INSTALLATION

We implemented a version of the Room as an interactive installation open to the public during fall 2023 (Figure 4). In this Section, we describe the implementation of the installation, showing how the framework described in Section 3 was used to guide the design.

Two users interact through the *digital filter* (Section 1.2): the Visitor enters the Room, which is the avatar of the Controller. Both experiences are in Virtual Reality, using Oculus Quest 2 headsets, communicating in real-time through a local network via UDP messages. As part of the experience, the Controller wears all four devices illustrated in Section 4.

The two participants, a Controller and a Visitor, were required to be present simultaneously. The installation aimed to explore the digital filter's ability to initiate interaction, so neither participant was informed that their experiences were linked. They were led to believe they were engaging in separate experiences to observe how they interacted with the environment and if they sensed the presence of "another".

Additional video material can be found in the Supplementary Material.

To create the installation, we needed to design:

- The Room body.
- The Room embodiment (control and perception).
- The interaction mechanism.

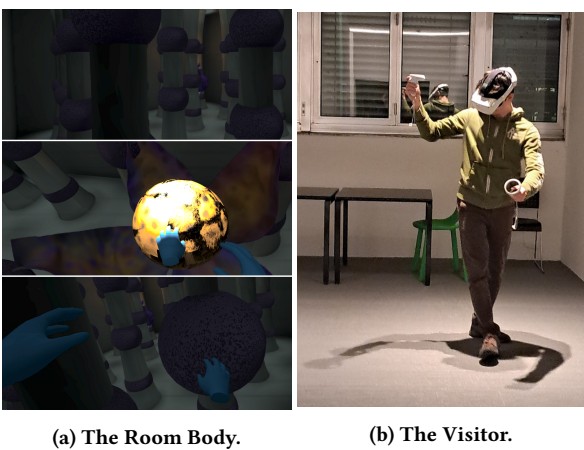

(a) The Room Body.     (b) The Visitor.

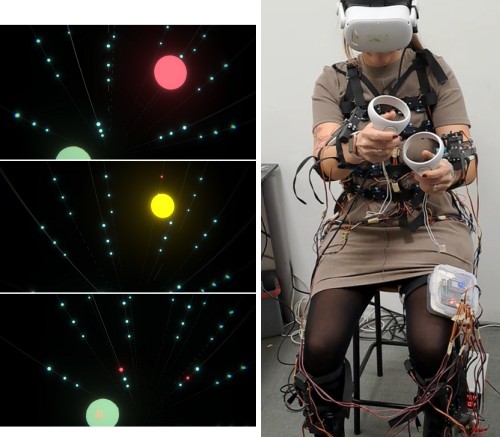

(c) The Controller view.     (d) The Controller.

**Figure 4: The views from the two communicating VR apps of the participants. In Figure 4c we have the Controller view. The Controller(Figure 4d) can touch, with its hands represented as yellow globes, the small spheres to inflate them, and touch the flower bud to open it. A red sphere indicates that the Visitor is touching the corresponding sphere on its side. In Figure 4a we have the Visitor view. The Visitor(Figure 4b) enters the digital room, with columns and the flower that the Controller can open. The purple spheres set in the columns are the elements that the Controller can inflate and that turn red on its side if touched by the Visitor. All these actions trigger the corresponding haptic devices worn by the Controller.**

These elements are interconnected, but the core decision was about the interaction mechanism, as the other two parts depend on it.

## 5.1 The Interaction Mechanism

We focused on the experience of the Controller to feel the motivations of the Room, supported by the haptic devices. Thus, as motivation for the Room, we chose *"persuasion of the Visitor"*, with added risk for the Controller (Section 3.3.1). Part of the Room's body is *"the flower"*, a flower bud that the Room can open to reveal a soft,

sensitive interior. Depending on the quality of the Visitor's touch on the flower, the Controller received two different sensations by the haptic devices: a gentle touch was translated into a pleasant sensation, and a rough touch in a displeasing one. Thus, the Controller needed to persuade the Visitor to touch the inside of the flower, while being sure that this touch was gentle. Our aim was that the perspective of the pleasing sensation motivates the Controller to try to communicate with the Visitor to persuade her/him, while the displeasing sensation should drive the Controller to establish a clearer communication to prevent the unwanted behaviour.

For the Visitor, we chose the basic motivation of exploration, which required the Room body to be itself an interesting space to discover (Section 3.3.2).

## 5.2 The Room Body

The first requirement of the Room was to contain *"the flower"* (Figure 4a), the focal element (Section 3.1.9). To attract the Visitor's attention, it needed to be spatially highlighted, obtained by placing the strongest light source inside it. To highlight it further, we composed the rest of the Room with a flock of elements (Section 3.1.1), contrasting its organic nature [36] to that of more geometric elements, with 16 fixed columns, each featuring 3 spheres that the Controller could independently inflate or deflate (Figure 4a). With this mechanism, the Room can create a variety of pathways, showing or hiding the flower itself if needed, which could anyway always be slightly perceivable [3], and also use the spheres as "reaching elements" to have haptic contact (Section 3.1.11) [19, 42]. These actions can be local or global, following the Visitor or attracting attention to specific locations (Section 3.1.8). Moreover, the spheres are the Room's main sensory organ, detecting the Visitor's touch.

## 5.3 Room Embodiment

The objective is to immerse the Controller, a human, into this completely different body. The Room avatar has the following possible actions:

- Inflating/deflating each sphere of every column.
- Opening/closing the petals.

And two sources of sensation:

- Whenever the Visitor touches any of the spheres.
- When the Visitor touches the inner part of the flower, either softly or roughly.

To be the new body, one's perspective must change with respect to it. No element is relevant but the ones that are part of the body, either as an action, or a sensation. Multi-sensory cues and visuomotor synchrony are enforced to foster embodiment (Section 2.1).

*5.3.1 The Sensory Translation System.* The Controller can only perceive what the Room avatar can do or perceive. To push our investigation to its limits, we chose to be as minimalist as possible. Visually, the Controller is immersed in darkness and can only see single spheres, which are gathered in groups of three, corresponding to the three spheres of the respective columns (Figure 4c). Spheres touched by the Visitor turn red on the Controller side, while the inside of the flower can either turn to a soft pleasing pink (soft touch) or deep red (rough touch) when the Visitor interacts with it. The Controller is actually seeing an abstract version of the Room

structure, much smaller, as big as her/his chest. The main source of feedback is haptic (Section 3.2.2). The Slider (Section 4.2.3) provides either pleasing or displeasing sensations based on the Visitor's touch of the flower core. Two sliders were used simultaneously, placed on the bare wrists, to maximise the haptic effect. The haptic vests (Section 4.2.1), either vibration or servo-actuated, with their 16 different locations, responds to touch of spheres on any specific column, providing a spatial coherence.

*5.3.2 The Control System.* To keep the system simple, we chose the easiest control mechanism, "puppeteer" (Section 3.2.1). The Controllers actuate the Room using their hands: they can inflate/deflate the spheres, and open/close the flower bud, by touching the virtual representations on their side. The control system is also integrated with haptic devices. Push (Section 4.2.4) and Tightening (Section 4.2.2) are used to obtain the PRoC effect [29], activating when the Controller makes the actions of opening/closing the flower and inflating/deflating the spheres, respectively.

## 6 EXPERIMENTS AND RESULTS

As in [76], we structured the installation as a laboratory experiment, to collect post-experience subjective reports. In this Section, we briefly describe our evaluation method and the main results we obtained.

## 6.1 Experiment and Questionnaire

The interactive installation features two parallel experiences (Controller and Visitor side, respectively), each lasting 5 to 7 minutes. Participants could engage in one experience at a time, unaware of their interconnection. Visitors were left to explore freely. However, [72, 80] argued that specific tasks enhance Sense of Embodiment (SoE), thus we assigned an objective to the Controller: to devise strategies to turn the flower bud pink, which actually depended on the Visitor's touch. A Google Forms survey was crafted to gather insights from 38 voluntary participants aged 16 to 66 (10 females, 28 males). Volunteers signed an informed consent to participate in anonymous form. The survey comprised two sections corresponding to each experience side. Common questions assessed enjoyment, immersion, perception of a living being, and allowed for free comments. Controller-specific questions addressed haptic device quality, wearability, comfort, and effectiveness.

The questionnaire and results are available in the Supplementary Material.

## 6.2 Results and Discussion

Overall, participants reported a very high enjoyment. They often reported the high level of immersion as one of the best features of both sides of the experience, and the need of discovery of both the environments.

On the Controller side, haptic devices received positive ratings for wearability, not impeding the experience or enjoyment despite covering the entire body. Moreover, participants found the haptic feedback to be one of the most intriguing aspect. However, setup time (3-4 minutes) was criticized, and some struggled to associate haptic sensations with visual cues and actions, possibly due to abstract visuals and slight response delays in the devices. This abstraction, coupled with the ambiguous objective, sometimes led

to frustration as some participants struggled to understand the effect of their actions or to find effective strategies.

On the Visitor side, participants reported a very high appreciation of the environment. The virtual ambient was interesting, a source of wonder, and fostered exploration, as we intended.

Some participants experienced a decline in engagement despite overall positive feedback. This was attributed to a "negative interaction loop": responsiveness of both virtual ambients heavily relies on the actions of the partner, and therefore sparse actions lead to decreased engagement, further reducing involvement and actions, creating a negative spiral.

Many participants struggled to perceive the presence of another living being in the environment. This difficulty is partially attributed to the "negative interaction loop". On the Controller side, participants were overwhelmed by stimuli from haptic and visual channels, making it challenging to interpret the abstract representation. On the Visitor side, the geometric and regular arrangement of space may have biased participants toward perceiving it primarily as a space rather than an organism.

Though they mostly did not understand they were interacting with another human, the incoherent, unpredictable behaviour of both environments sparked discovery and also play. On the Controller side the environment, entirely abstract, posed as a challenging puzzle to be deciphered and solved, while on the Visitor side it offered an intriguing space for exploration. For those who sensed the presence of another being or understood the true setup, play often resembled games like hide and seek or tag, with the Visitor attempting to touch spheres before the Room could inflate them or the Room seeking out the Visitor with the spheres. These users reported the highest levels of enjoyment, indicating how the setup can serve as a tool for interaction and stimulate creative behaviors.

The most positive result was on perspective change. Upon learning or understanding the true nature of the experience, they were surprised and amazed by the opportunity to "become" a different being and interact with a radically new entity. Moreover, they appreciated the novel way of communicating with another human being, which required both parties to adapt and leverage their bodies differently.

Overall, these findings underline the importance of finding a compromise between the stimulating abstraction and the necessary intelligibility of the environment, and confirm the quality of the system in its ability to challenge, engage and surprise users.

## 7 CONCLUSIONS

We embarked on the exploration of spaces as non-anthropomorphic bodies. With the general objective of studying how we humans can embody these new aliens, perceive them as alive and being able to communicate with them non verbally, we chose to investigate spaces, for how they radically challenge human morphology and interaction mechanisms we are used to. In our setup, two humans interact through a *digital filter*. The Controller embodies the Room, the space as the non-anthropomorphic avatar, and the Visitor enters and interacts with the Room.

Our primary contribution is an extensive review of existing research on embodiment, interactive exhibition design, and interactive installations, leading to the development of a comprehensive framework for designing our system. This framework addresses the challenges of designing the Room as a living organic being, considerations regarding embodiment (perception and control) of the space, and motivations for interaction between these different entities. Additionally, we created four custom haptic devices to convey a range of sensations, essential for integrating sensory input into the Controller's experience to match the distributed nature of the Room's body. Furthermore, we present an initial implementation of the Room system guided by our framework, an interactive VR art installation, which also served as an experimental context for our study.

Results were very promising. Immersion and enjoyment were very high.

Humans are less sensitive to tactile input compared to visual stimuli, and they are not accustomed to primarily receiving stimulation through the haptic channel. To address this, we developed haptic devices that offer sensations as diverse as possible. These devices obtained positive scores for wearability, comfort and immersion, though the stimulation they provided could not always be linked to the experience due to small lags and the abstract nature of the experience.

A crucial aspect is the clarity of the experience, especially for the Controller, immersed in a completely abstract environment. Participants often reported a need to be more guided, and lack of understanding often hindered the possibility to feel the presence of the other being in the environment.

Participants often showed wonder and playfulness, using and imagining the system as a source of creative behaviour and novel forms of interaction.

### 7.1 Future Directions

With the theoretical framework largely developed, now the project needs to focus on implementations capable of obtaining the desired results and thoroughly test our hypotheses. The approach we will follow is to acquire evidence by objective methods, such as logs, observations and qualitative utterances. The next steps will focus on the following aspects:

*7.1.1 Clarity on the Controller Side.* Finding a compromise between the desire for minimal visual cues and for the Controller to be able to understand its context. To do this, we need to expand the framework with more specific principles for the Controller perception, while also improving the haptic devices to solve the technical lag issues. Additionally, reducing Controller stimulation may enhance focus on relevant aspects.

*7.1.2 Physical Room.* Interactions with spaces are predominantly haptic, which VR on the Visitor side couldn't fully replicate, resulting in limited feedback when touching elements. To address these limitations, future Rooms will be physical, mechatronic spaces, aligning better with the framework's principles. While the initial VR exploration facilitated data collection with a faster implementation, physical spaces will enable us to maximize the framework's potential, allowing for genuine haptic interaction, animated components, and organic structures and materials, thus ensuring a comprehensive and coherent experience.

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
