# OpenReview forum: "The Room: design and embodiment of spaces as social beings"
_acmmm.org/ACMMM/2024/Conference — MM2024 Poster_

### Official Review · Reviewer_gCrm · 2024-05-22

**Rating:** 4
**Confidence:** 4

**Summary:**

The paper addresses the investigation of spaces as non-anthropomorphic avatars. Based on the discussion of avatar embodiment and interaction with space, a pilot framework for Room design in VR has been implemented, addressing challenges related to its body, perception, and interaction process. The related real-time interactive installation takes the form of a two-player environment, where one player is active as controller of the room and the other represents the visitor.  The main interaction/perception method is haptic. An evaluation of the experiment has been provided.

**Strengths:**

Considering the ongoing discussions with respect to the metaverse or adaptive real-time physical environments, the paper addresses a timely and relevant topic, namely what does it mean if space, in form of a room, is not considered as a non-anthropomorphic unit but rather becomes an entity that perceives mainly through tactile sensations and is then driven by goals and desires.

The contextualisation of the problem in form of a literature survey is detailed with good resources.

The outlined framework focuses on a simple interaction between room and one interactor in a VR environment. This simplification is valid and relates well to the provided implementation.

The approach of the implementation is sound, and the provided engineering is appreciated.

An experimental validation of the implemented VR environment has been provided, mainly testing the experience of controller and visitor independently but also investigating the understanding of participants with respect to the role of the other.

Overall, the paper is well-written.

**Limitations:**

The paper has two problems that ned to be addressed if it gets accepted.

First, there is a misalignment between the aims and goals outlined in sections 1 to 3, and the then presented implementation and related experiment and findings. The overall vision of the authors aims to build environments that represent embodied rooms.  They explained all this in very much detail. Yet, the design framework is much simpler by providing three parts: room, embodiment and interaction. The various sub elements within those parts cover a very limited set of what had been outlined earlier. Why this is has not been explained and that causes a first misalignment between potential uses (which also have never been mentioned) and the actual current work. The framework is used to build a first prototype but that is even simpler in scope. There is one room, it is static in space expansion, provides a limited variety of communication agents as self-representation (2 – the flower and pillars) and facilitates the interaction with one visitor at a time. In relation to that the tasks it can ask to be performed by the visitor are limited to exploration in general, and the flower in particular and the means of reactions of the controller are fitted to this task. This is not a problem if the grand statements with respect to generalisation had not been made and simply the findings had been reported. In this context of misalignment, the conclusion is another instance. Here the authors use the findings of the experiment as the basis to explain why their next step of development is a real physical environment. This is problematic as the experiment has been performed in VR, that offers ways of handling space not available in the real world. Thus, the authors would first have to investigate materials that would facilitate the experience of expanding or shrinking space. In this context they would also have to first investigate what happens if those environments cross physical constraints that are dangerous for human beings, such as performed pressure on them or to cold or hot environments.  Finally, As the room in VR had been controlled by a human the authors first had to explore how far the human cognition can be replaced by an agent so that at the end the room is embodied. This can not yet be concluded by the findings of the experiment. Thus, this aim of the future work should be moved as the final goal and not so prominent as it is placed now.

The second problem, and much easier to be fixed, is the vagueness of the questionnaire setup, and the presentation and discussion of the findings. It had been appreciated if the authors provided a bit more information about the Google form they created. How many questions have been asked for each of the question categories (enjoyment, immersion, perception of a living being), how many of those have been free text and how many used what types of quantitative measurement. A similar detail would be good for the controller part of the questionnaire (here it seems that all values are based on a Likert scale). Regarding the collection of the test population: the authors mention that the implementation was made accessible during fall 2023 (see section 5) but that the experiment was designed as a lab-based experiment (section 6). Please clarify is both refer to the same event and also state how the participants have been invited. A side note: no supplementary material has been accessible for the reviewer. Suggestion: always provide the essential information in the text or at least provide an anonymised link in the paper so that a reviewer or any other reader can check. A bit more information on the demographics, i.e. experience with VR) had been good. A short statement about the type of coding performed on free-text answers is necessary.

Considering all the collected data it is astonishing that the authors described findings and discussed them in statements like this (comments in <> by the reviewer: “ Overall <How many exactly and what did the other state?> , participants reported a very high enjoyment <Very high means what exactly?> . They often <how often?> reported the high level of immersion as one of the best features <what other features had been mentioned and is there a ranking?> of both sides of the experience, and the need of discovery of both the environments. <it seems that you men “on both sides”? If not then the authors have to state earlier that all 38 participants played both roles>”.  Another example is : “On the Controller side, haptic devices <all of them received the same value? If not a table with numbers would be great>received positive ratings
for wearability <on what scale?>, not impeding the experience or enjoyment despitecovering the entire body <this statement was done by which of the participants and if not by one only by how many then?>.
Thus, the authors are strongly advised to use the date (averages, median, standard deviation) as collected to provide the reader with an insight what has been measured. If there are opposing meanings those need to be contextualised with verbatim references.
Finally, the section on “perception change” is interesting. However, it is unclear if the controller and visitors performed that by themselves or received a hint AND it would be good to know how many of each group achieved that level and in what time.

**Suitability:**

3

---

### Official Review · Reviewer_RcgR · 2024-05-23

**Rating:** 3
**Confidence:** 4

**Summary:**

This study examines the investigation of places as non-human representations. They are studying the interaction with creatures that possess characteristics distinct from those of humans in order to comprehend how they might be represented as avatars and seen as alive, social entities. In order to fully explore this topic, they have created an interactive area called the Room, which pushes the boundaries of both human-like structure and common social interaction processes. They provide a preliminary framework for designing the Room, which aims to tackle the difficulties associated with its physical structure, sensory perception, and interactive procedures. They have developed a real-time, two-player, virtual reality experience that serves as an interactive installation. This experience revolves around the Room avatar and places a strong emphasis on haptic input as the primary method of perception for the participant inhabiting the Room.  By explore the fundamental elements of physical presence and social understanding, this work may open a new niche in VR's interaction techniques.

**Strengths:**

+ The work has an interesting device, with detailed designs (about first six pages).
+ In general, the writing is easy-to-understand.

**Limitations:**

- Unfortantely, the result section is extremely short and more results are expected.
- some organization issue, e.g., line 882-883, one-sentence paragraph.

Considering the workdone and the details of the gadget, it is recommended to consider this work in Work-in-Progress or Demo (If we have).

**Suitability:**

2

---

### Official Review · Reviewer_Xokj · 2024-06-06

**Rating:** 3
**Confidence:** 2

**Summary:**

The paper attempts to explore regarding virtual spaces as “non-anthropomorphic avatars.” The authors seem to have developed a virtual reality and haptic-based system to evaluate the suggested framework.

**Strengths:**

The paper seems to explore an interesting research theme regarding the environment in virtual reality (VR). With VR's increasing adoption, I see potential benefits of this kind of work, especially in regard to creating interactions.

**Limitations:**

There are multiple limitations in the paper.
1)	Introduction: I think the introduction needs to be expanded further. First, it does not establish the background and context of the research, particularly describing what is meant by “non-anthropomorphic avatars” and “anthropomorphic” based on current works (maybe consider proving examples based on contemporary literature) and how this research fits the current domain of Multi-Media (MM) and Human-Computer Interaction (HCI). I think part of this is described in “Relevance to the Conference” but needs to be further expanded in the Introduction section. Second, the introduction section lacks emphasizing the novelty of this work; particularly, why this work is important and how this is different from other related work. Authors have included related work in Section 2, but I suggest describing the importance of this work and how it's different from contemporary literature based on a critical evaluation of previous work.

2)	Experiment and Questionnaire: Experiments seem to have involved participants under 18 years old. The paper doesn’t mention whether the research has been approved by any regulatory organizations like the Institutional Review Board (IRB). This can raise questions regarding research ethics since the age of consent is usually 18, though regulations regarding the age of consent may vary depending on the region. Therefore, it is recommended that the authors provide more details on this, especially regarding whether the authors followed any applicable regulations and got approval from relevant regulatory bodies in the institution (e.g., IRB).

3)	Results and Discussion: The results of this work seem vague. First, the paper lacks a cohesive framework for evaluating the proposed system. The paper mentions the questionnaire as supplementary materials in line 794; I apologize, but I didn’t find this in the paper or my review invitation. The authors mention, "Common questions assessed enjoyment, immersion, perception of a living being, and allowed for free comments.". But this seems vague and lacks specific details about questions. Therefore, it would be great if the authors mentioned what questions were exactly asked and any type of scale (e.g., Likert) used, justifying why such questions were asked of participants. Second, the paper lacks statistical evaluation. Several statements, such as the ones in lines 800, 815, and 824, seem unsubstantiated without data to back them up. I suggest including a statistical evaluation and relevant figures that describe the results.

4)	Minor comments:
       a) Line 7 – word alignment issue
       b) The first letter of each word in the title should be upper case (except words like ‘and’, ‘of’ and ‘as’)

**Suitability:**

2

---

### Official Review · Reviewer_aztf · 2024-06-06

**Rating:** 6
**Confidence:** 3

**Summary:**

This paper investigates how entities with non-human features can be embodied and perceived as living, social beings. The study introduces a pilot framework for the design of an interactive space, referred to as "the Room," which challenges traditional notions of anthropomorphic structures and social interaction mechanisms. This framework is implemented in a real-time, two-player virtual reality (VR) experience where one participant embodies the Room, and the other interacts with it, focusing on haptic feedback as the main mode of perception.

**Strengths:**

The paper presents a novel research project focused on "non-anthropomorphic avatars" that explores the design and implementation of interactive spaces which act as avatars. The system developed provides users with a real-time, two-player VR experience, which is also used to gather data. An innovative aspect of the research is the integration of haptic devices into the VR environment, emphasizing sensations, wearability, and implementation to create a comprehensive sensory translation system. This paper presents a pilot study demonstrating how multimodal interfaces (haptics and VR) can be leveraged to embody and interact with non-anthropomorphic avatars, particularly spaces. The theoretical foundation is robust, pushing the boundaries of embodiment and social cognition research. The paper's structured presentation, detailed VR setup, and thorough background exploration enhance its clarity and impact. These applications extend beyond academia, potentially influencing fields like social robotics and entertainment by providing new ways to design interactive, empathetic non-human characters.

**Limitations:**

One limitation of the paper is its pilot nature, which means the findings are preliminary and may not be generalizable without further validation. The scope of the evaluation is somewhat limited, as it primarily relies on a single interactive installation to gather data. Additionally, the lack of comparisons with existing methods or frameworks in the field could be seen as a missed opportunity to contextualize the findings within the broader landscape of embodiment research. However, this paper is still overall good and should be a good paper for the MM community.

**Suitability:**

3

---

### Meta-Review · Area_Chair_LQPS · 2024-07-02

**Recommendation:** Accept (Poster)
**Confidence:** 4

**Metareview:**

This paper investigates how entities with non-human features can be embodied and perceived as living social beings. The study introduces a pilot framework for designing an interactive space, referred to as "the Room," which challenges traditional notions of anthropomorphic structures and social interaction mechanisms. This framework is implemented in a real-time, two-player virtual reality (VR) experience where one participant embodies the Room, and the other interacts with it, focusing on haptic feedback as the main mode of perception.

This paper presents a compelling and innovative approach to exploring non-anthropomorphic avatars in VR, mainly focusing on haptic feedback. The paper is well-structured and theoretically robust, offering significant contributions to the field. However, the preliminary nature of the findings, limited evaluation scope, ethical considerations, and insufficiently detailed results and discussion sections are notable limitations.

While Reviewer aztf strongly supports acceptance, reviewers Xokj and RcgR express concerns about the paper's current state, leaning towards rejection but open to persuasion. Reviewer gCrm provides a balanced view, leaning towards acceptance with the expectation that the final paper will address the highlighted issues.

Considering the innovative nature of the research, its potential impact, and the detailed rebuttal addressing reviewers' concerns, I recommend an acceptance as a poster presentation. The detailed feedback will help the authors to address the noted limitations in a future version of this paper.